# Evaporation Caused Invaginations of Acoustically Levitated Colloidal Droplets

**DOI:** 10.3390/nano13010133

**Published:** 2022-12-27

**Authors:** Hongyue Chen, Yongjian Zhang, Heyi Wang, Xin Dong, Duyang Zang

**Affiliations:** 1MOE Key Laboratory of Material Physics and Chemistry under Extraordinary Conditions, School of Physical Science and Technology, Northwestern Polytechnical University, Xi’an 710129, China; 2Shaanxi Key Laboratory of Surface Engineering and Remanufacturing, School of Mechanical and Material Engineering, Xi’an University, Xi’an 710065, China

**Keywords:** colloidal droplet evaporation, acoustic levitation, evaporation flux, surface invagination

## Abstract

Controlled buckling of colloidal droplets via acoustic levitation plays an important role in pharmaceutical, coating, and material self-assembly. In this study, the evaporation process of PTFE colloidal droplets with two particle concentrations (60 wt% and 20 wt%) was investigated under acoustic levitation. We report the occurrence of surface invagination caused by evaporation. For the high particle concentration droplet, the upper surface was invaginated, eventually forming a bowl-shaped structure. While for the low particle concentration droplet, both the upper and lower surfaces of the droplet were invaginated, resulting in a doughnut-like structure. For the acoustically levitated oblate spherical droplet, the dispersant loss at the equatorial area of the droplet is greater than that at the two poles. Therefore, the thickness of the solid shell on the surface of the droplet was not uniform, resulting in invagination at the weaker pole area. Moreover, once the droplet surface was buckling, the hollow cavity on the droplet surface would absorb the sound energy and results in strong positive acoustic radiation pressure at bottom of the invagination, thus further prompting the invagination process.

## 1. Introduction

Evaporation of colloidal droplets is ubiquitous in nature and is also widely used in a variety of industrial and pharmaceutical processes [1]. The colloidal droplets evaporating on solid walls would lead to the formation of various patterns, ranging from the typical “coffee ring” to radial, circular, and craquelures crack patterns [2,3,4]. Evaporation is used to arrange the colloidal particles into desired structures, which became a new approach for material self-assembly [5,6]. Therefore, the evaporation of sessile colloidal droplets has important applications in coating [7], inkjet printing [8], and the preparation of nanocomponents [9]. Moreover, due to the loss of dispersant and interlocking between particles, the droplet undergoes a sol–gel transition, during which the particles aggregate to form an elastic shell [10]. Therefore, the evaporation of colloidal droplets does not simply shrink isotropically; however, they may undergo significant mechanical instability, such as surface buckling and fracture [11,12,13].

It has to be noted that during the evaporation of a colloidal droplet based on a solid surface, different factors including surface properties [14,15], the pinning effect of the contact line [16], and the heat transfer at the solid-liquid interface [17] affect the interaction between particles and the final dried pattern. Therefore, it was desirable to avoid contact between droplets and solid surfaces. Spray drying [18,19] is one of the convenient and contact-free methods which has been widely used for evaporation-induced particle self-assembly. Atomized colloidal droplets would form hollow or curved particle accumulation structures after drying, which could be widely used in food, pharmaceutical, ceramic processing, and the preparation of new functional microstructures. However, it is hard to obtain the overall information on the evaporation process for a single droplet during spray drying.

Compared to spray drying, the acoustic levitation technology uses acoustic radiation force to levitate the droplet at the sound pressure node, which completely avoids the influence of the solid–liquid interface during the evaporation process [20]. Moreover, it is convenient to observe the evaporation process of a single droplet in real-time, as well as manipulate the structure of the evaporation-induced self-assembly [21,22]. Under acoustic levitation, surface buckling during evaporation was observed, which is the same as a lot of experimental phenomena in spray drying. The buckling dynamics of drying colloidal droplets is of particular interest [13,18,23,24,25]. Generally, the surface buckling is attributed to the capillary pressure in the menisci at the interstices between the particles in the elastic shell. The capillary pressure induced the instability of the shell and led to buckling of the surface to relieve the strain, which further causes the shell to deform. Buckling occurs first at the weak area of the elastic shell, where the mechanical strength of the shell is too weak to resist the capillary pressure.

Many studies have found that the spatially varying thickness of the elastic shell formed on the acoustically levitated droplet surface after the gelation transition is not uniform, which is mainly manifested in the following two levels. (1) The thickness of the top of the droplet is smaller than that of the bottom. This is due to the density stratification caused by the asymmetric flow structure inside the droplet and the effect of gravity [26]. (2) The equatorial area of the droplet was thicker than the pole area. As a result, buckling occurs in the two pole areas of the droplet or only in the north pole area, resulting in a dry doughnut-like or bowl-like structure [27,28]. However, current studies have not fully explained the formation reasons for the thickness distribution in the second case, especially the mechanism of evaporation. There is still a lack of a full understanding of the effect of the sound field on the invagination.

Herein, we studied the morphology evolution of PTFE colloidal droplets with varied particle concentrations during evaporation under acoustic levitation. It was found that, for the droplet with a high particle concentration (60 wt%), under the induction of evaporation, the upper surface of the droplet was invaginated to form a bowl structure. While for a low particle concentration (20 wt%), the upper and lower surfaces of the droplet were both invaginated to form a doughnut-like structure. We focus on the distribution of the evaporation flux on the oblate spherical droplet under acoustic levitation and aim to illustrate the formation mechanism of the nonuniform thickness of the elastic shell on the droplet surface. In addition, the acoustic radiation pressure of different structures was calculated and the mechanism of the evaporation-induced surface invagination was further explained from the mechanical perspective.

## 2. Experiments and Materials

### 2.1. Materials

The colloidal dispersion used in the experiment was diluted by the originally concentrated dispersion with a polytetrafluoroethylene (PTFE) particle content of 60 wt% to obtain colloidal dispersion with a lower particle concentration (20 wt%). The detailed properties of the PTFE condensed dispersion are given in Table 1. Before the experiment, the colloidal dispersion was sonicated for 30 min using an ultrasonic probe (Bilon-650Y, Shanghai, China) to avoid particle aggregation.

### 2.2. Experimental Setup and Methods

In this study, the levitation of colloidal droplets was realized using a uniaxial acoustic levitator and the schematic illustration of the experimental apparatus is shown in Figure 1. The acoustic levitator (SonoRh-1, Shengli Ltd., Nangjing, China) consisted of an emitter and a reflector arranged coaxially along the gravitational direction. The emitter emitted a sinusoidal ultrasonic wave with a frequency of 20.5 kHz, which was reflected through the concave reflector, forming a standing wave sound field between them. The sample droplet was manually injected into one of the sound pressure nodes in the levitator using a microsyringe. To record the morphology evolution of the levitated droplet, the process of the droplet evaporation was captured by a high-definition CCD video microscope (GP-640S, GAOPIN, Shenzhen, China) in real-time. Using the recorded images, we then used MATLAB R2014a with an in-house compiled code to extract the droplet’s contour and obtain the droplet width and height as a function of the evaporation time. To obtain more information about the morphology and structure of evaporative deposition, a hand-held digital microscope (HOT, HT60S, Shenzhen, China) was used to observe the top and vertical sections of the deposition products.

### 2.3. Sound-Field Simulation

The acoustic radiation pressure on the sample droplet surface at the initial shape of a stable levitated state was calculated using the commercial finite element software COMSOL Multiphysics 5.6. For calculation, a three-dimensional model was used to establish the simulation domain which corresponded with the geometry of the levitator. The simulation domain was divided using a tetrahedral mesh and the user’s predefined size was ultrafine. The rigid boundary condition was used at the emitter and the reflector and the sidewall was set as cylindrical wave radiation. The boundary acoustic pressure was restrained by the Helmholtz equation:(1)$$\nabla \cdot \left(-\frac{1}{\rho_{0}}\nabla p\right)-\frac{\omega^{2}p}{\rho_{0}c_{0}^{2}}=0$$
where *p* is the sound pressure, $\rho_{0}$ (= 1.18 kg/m^3^) is the density of air, ω is the angular frequency, and $c_{0}$ (=346.12 m/s) is the speed of sound in air.

The initial condition of the calculation was obtained from the amplitude of the emitter:(2)$$n\cdot \left[\frac{1}{\rho_{0}}\left(\nabla p\right)\right]=a_{0}$$
where *a*_0_ (=0.79 × 10^6^ m/s^2^) was the normal acceleration of the emitter.

After obtaining the sound pressure distribution of the sound field, the acoustic radiation pressure on the droplet surface *P*_A_ could be calculated by King’s theory [29]:(3)$$P_{A}=\frac{1}{2\rho_{0}c_{0}^{2}}p^{2}-\frac{1}{2}\rho_{0}v^{2}$$
where *v* is the velocity of the medium. The angular brackets in Equation (3) denote the time average over one period of acoustic oscillation. For simplicity, the material of the droplet was assumed to be water (*ρ* = 998.2 kg/m^3^ and the sound velocity of water *c*_water_ = 1495.33 m/s) because the acoustic impedance mismatch between air and all of the colloidal solution used in the experiments was similar.

## 3. Results and Discussion

### 3.1. Morphology Evolution during Evaporation

Upon evaporation, the morphologies of the acoustically levitated droplets evolved with increasing evaporation time, which can be evidenced from the side-view photographs (Figure 2B,C). In order to obtain the optimized levitation stability of the droplet, preventing the droplet from atomizing under a sound intensity that was too high or falling under a sound intensity that was too low, the initial shape (*t* = 0 min) of the two concentration droplets was preset to be a spherical oblate by adjusting the emitter–reflector distance. The static oblate shape of the levitated droplet was determined by the acoustic radiation pressure exerted on the droplet surface *P*_A_. As shown in Figure 2A, *P*_A_ was not uniformly distributed on the droplet surface. It was positive (compression effect) at the pole area but negative (suction effect) at the equatorial area, which was in line with our previous results and others [30,31].

For different particle concentrations of droplets with the same initial shape, we observed the morphology evolution during the evaporation process. For droplets with a high particle concentration (60 wt%) (Figure 2A), it was observed that the upper surface of the droplet gradually became flatter and the final side view showed a droplet shaped like a bowl. For droplets with a low particle concentration (20 wt%), as the droplet shrunk, it almost always kept its oblate shape until it finally became a cake-like shape.

Furthermore, the variation of droplet size with time was quantitively analyzed (Figure 3). Clearly, the evaporation time of a low particle concentration droplet increased significantly due to the low content of solid particles and the high content of water. For the 60 wt% droplets, the length of the major axis decreased significantly in the first 10 min of evaporation and then it almost stayed stable. The change in the length of the minor axis of the droplet was not obvious. The 20 wt% droplet’s major axis length also shrunk rapidly during the initial period of evaporation, while the droplet’s minor axis length remained almost constant. After 25 min, the 20 wt% droplet’s major axis length decreased slowly, while the minor axis length decreased faster. It could be observed that the variation trend of the major and minor axes was the same for the two particle concentrations. The major axis decreased rapidly at the initial period of evaporation and then tended to remain stable, while the minor axis had the opposite trend. It was supposed that the dispersant evaporation caused the droplet to become smaller at the initial period of evaporation. With further evaporation, the particles agglomerated at the air–liquid interface to form an elastic shell structure [28], which slowed down the evaporation rate.

To obtain more details of the structure of the dried colloidal droplets, the top and vertical sections of the dried products were photographed (Figure 4). Compared with the side view, it could be seen that the flattened surface of the droplet in the side view actually had an invagination in the center. Specifically, the upper surface of the droplet with a high particle concentration (60 wt%) was invaginated inward and finally formed a bowl-like structure. In contrast, the 20 wt% droplet had both surfaces invaginated inward and it was worth noting that the upper surface was invaginated deeper than the lower face.

### 3.2. Formation of the Gelation Shell

With the evaporation of the dispersant, the colloidal droplet first shrunk uniformly with an in situ increase in particle concentration at the droplet’s surface [32]. With further evaporation, Basu et al. and Saha et al. [26] pointed out that there was a high value of Peclet numbers (*Pe* = *t_diffusion_*/*t_evap_*, and 5 < *Pe* < 250) for evaporating levitated colloidal droplets. This means that the diffusion of particles was inadequate to mitigate the inhomogeneity in particle distribution throughout the droplet. Therefore, particles would accumulate at the gas–liquid surface to form a dense elastic shell. However, the thickness of the shell formed due to evaporation varies widely along the droplet interface. Miglani and Basu [28] provided SEM images demonstrating that the equatorial area of dried colloidal droplets has greater thickness relative to the poles. Herein, as it was proved before, the length of the droplet’s major and minor axis was nonlinear with evaporation time. To gain deep insight into this phenomenon, we qualitatively analyzed the distribution of the local evaporation rate on the droplet surface. A large number of studies have elucidated that the boundary layer acoustic streaming around the levitated droplet could significantly improve the mass transfer capacity during evaporation [33,34,35], which is manifested by the increase in the Sherwood number (*Sh*). For the oblate spherical droplet under acoustic levitation, the mass transfer capacity on the droplet’s surface was not uniform due to the influence of the boundary layer acoustic streaming and the spatial dependence surface curvature. The local *Sh* number over the droplet surface reads as [36]:(4)$$\langle Sh\rangle =2{\left(\frac{45}{4\pi }\right)}^{0.5}\frac{B}{{\left(\omega D_{0}\right)}^{0.5}}\frac{sin^{2}\left(x/a\right)}{{\left(1+sin^{2}\left(x/a\right)\right)}^{0.5}}\;$$
where *B* = *A*_0e_/*ρ*_0_*c*_0_ is a velocity scale if the incident sound wave that depends on the sound pressure amplitude *A*_0e_, *ω* is the angular frequency of the incident sound wave, *D*_0_ is the diffusion coefficient, *x* is the arclength reckoned from the point *O*_1_ (Figure 5) to point *O*_2_, and *a* is the volume equivalent radius. The angular bracket in the equation denotes the time average over one period of acoustic oscillation.

Equation (4) shows that the time average mass transfer rate on the surface of the droplet is at its maximum at the equatorial area and decreases along the *x*-direction toward the pole. Obviously, the larger the mass transfer rate is, the more the dispersant mass is lost by evaporation; that is, the larger the evaporation rate is at this location. According to this, the distribution diagram of the evaporation rate on the droplet surface could be drawn in Figure 5, where the length of the arrow qualitatively represents the magnitude of the evaporation rate. This further indicates that, due to the high evaporation rate at the equatorial area, where the evaporation amount of the dispersant was larger, the length of the major axis decreased rapidly in the initial period of droplet evaporation. The non-uniform evaporation rate on the droplet surface, on the one hand, led to the preferential aggregation of particles in the equatorial area. On the other hand, in order to supplement the loss of solvent here, there may also be a flow from the pole to the equator inside the droplet, which further carried more colloidal particles to agglomerate at the equatorial area. As a result, the droplet surface formed an elastic shell that gradually thins from the equatorial area to the poles.

### 3.3. Mechanism for the Invagination

The static shape of the acoustically levitated droplet is determined by the balance between acoustic radiation pressure *P*_A_, surface tension *σ,* and internal pressure *P*_i_, which can be described by [31]$\;P_{i}-P_{A}=\sigma \nabla \cdot n\;$(where *P*_A_ is a time averaged pressure caused by ultrasound, **n** is the normal unit vector on the droplet surface pointing outward, and ∇·n is the total local curvature). The shape evolution during evaporation is determined by the competition between these forces. Upon evaporation, colloidal particles accumulate at the liquid–water interface [1], leading to the formation of a gel shell. Due to the inhomogeneity of the evaporation flux, the shell has a non-uniform thickness. Once the acoustic compression together with the enhancing negative capillary pressure exceeds the mechanical strength of the gel shell, buckling occurs [37]. It should be noted that the pole areas have the thinnest thickness and thus the lowest mechanical strength to resist the depression caused by an acoustic radiation force. Therefore, surface buckling first appeared at the pole area.

As previously reported, the cavity structure in the sound field has a strong absorption of sound energy [38]. To understand the mechanical origin of invagination, the acoustic radiation pressure distribution in the sound field and on the surface of the dried product of the colloidal droplet with different morphology was calculated (Figure 6). As shown in Figure 6A,C, there was strong positive pressure inside the cavity, indicating that strong sound energy absorption was caused by the invagination. Therefore, once the droplet surface buckled due to instability, the absorption of sound energy by the cavity promoted the invagination and, in turn, reinforces the sound energy absorption. In other words, acoustic radiation force provides positive feedback for the shell invagination as well.

The acoustic radiation pressure on the upper and lower surfaces of the droplet along the radial direction was derived (Figure 6B,D). The distribution of acoustic radiation pressure on the upper surface is slightly greater than that on the lower surface. The larger inward positive pressure made the upper surface of the droplet invaginate deeper. Moreover, under the effect of gravity, particles tend to deposit at the bottom of the droplet [28]. For droplets with a high particle concentration (60 wt%), there were enough particles to accumulate at the bottom of the droplet due to stronger gravitational sedimentation, leading to the formation of a thicker elastic shell at the bottom to resist the acoustic radiation pressure. At the top of the drying droplet, however, the shell is weak due to both slow evaporation and gravitational sedimentation. As a result, only downward invagination at the top surface occurred and led to a bowl-shaped morphology. For 20 wt% droplets, a large number of particles accumulated at the equatorial area under the action of evaporation and internal flow and fewer particles could be deposited at the bottom. Therefore, both the bottom and top surfaces were soft and unable to resist the compression caused by capillary pressure and acoustic radiation force, and eventually resulted in both upward and downward invagination, which is responsible for the formation of doughnut-shaped morphology.

The results indicate gravity still plays a role in the evaporation of colloidal droplets, in which sedimentation is hard to be avoided through the effect of solid–surface contact. which has been suppressed. The shape evolution is a competition between acoustic radiation force and the mechanical parts of the shell, which both change with time. The different invagination manners, downward or both downward and upward, resulted in the final bowl-shaped or doughnut-shaped morphology.

## 4. Conclusions

In this study, we experimentally investigated the evaporation process of PTFE droplets under acoustic levitation. It was found that the final dried morphologies varied on the particle concentration of the droplets. For droplets with a high particle concentration (60 wt%), there were enough particles to accumulate at the bottom of the droplet due to stronger gravitational sedimentation, leading to the formation of a thicker elastic shell at the bottom to resist the acoustic radiation pressure. As a result, only downward invagination at the top surface occurred and led to a bowl-shaped morphology. For droplets with a low particle concentration (20 wt%), insufficient particles could be deposited at the bottom. Therefore, both the bottom and top surfaces were soft and unable to resist the compression caused by capillary pressure and acoustic radiation force, and eventually resulted in both upward and downward invagination, which is responsible for the formation of doughnut-shaped morphology. These results may provide a reference for the development of more controlled surface buckling techniques for colloidal droplets.

## Figures and Tables

**Figure 1 nanomaterials-13-00133-f001:**
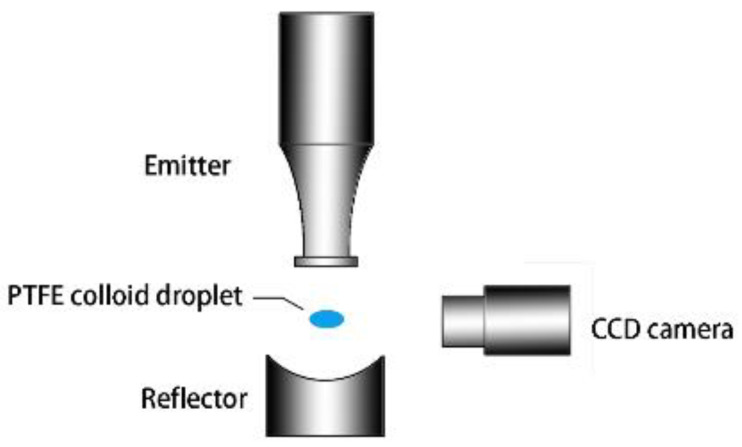
Schematic diagram of the experimental apparatus.

**Figure 2 nanomaterials-13-00133-f002:**
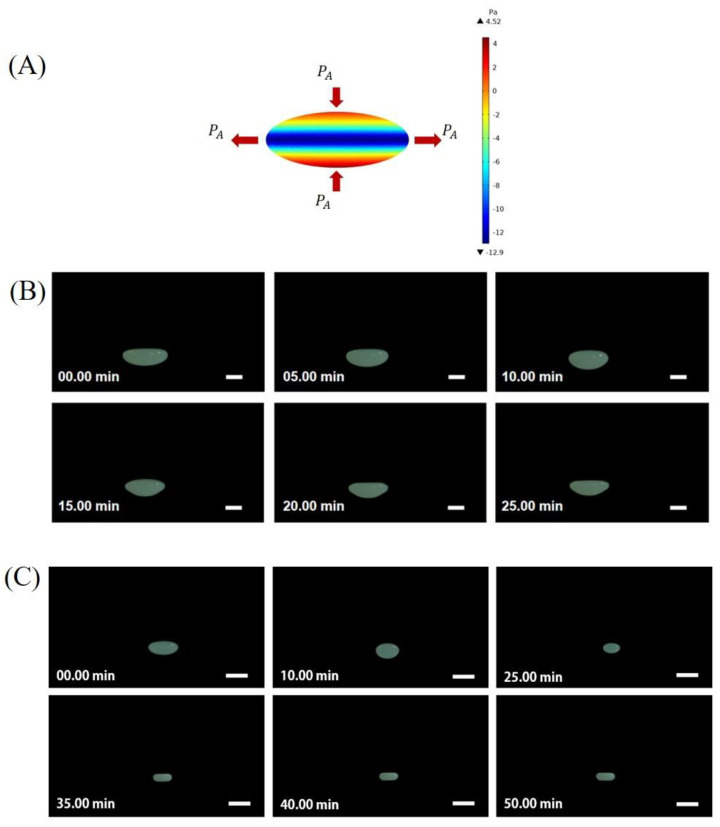
Morphologic evolution recorded from the side view of acoustically levitated PTFE droplets during evaporation. (**A**) Acoustic radiation pressure on the surface of an acoustically levitated droplet with an oblate initial shape. (**B**) The evaporation process of 60 wt% droplets. (**C**) The evaporation process is 20 wt%. All scale bars are 1 mm.

**Figure 3 nanomaterials-13-00133-f003:**
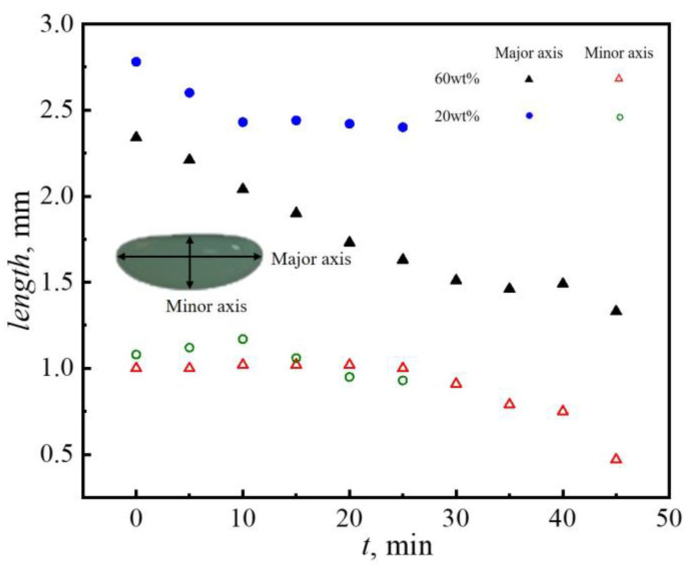
Variation of the length of the major and minor axes of acoustically levitated PTFE colloidal droplets with the evaporation time.

**Figure 4 nanomaterials-13-00133-f004:**
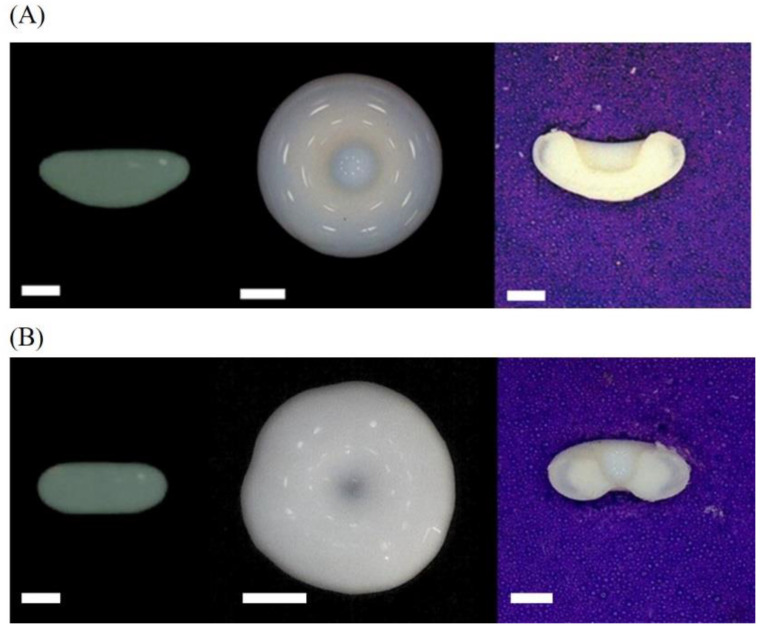
The morphology of the final relic for acoustically levitated PTFE droplet (**A**) The 60 wt% droplet invaginated from the upper surface. (**B**) The 20 wt% droplet invagination from both the upper and lower surfaces. The three images from left to right are the side view taken with a high-definition CCD camera, the top view taken with a hand-held microscope, and the vertical section image taken with a microscope. All scale bars are 1 mm.

**Figure 5 nanomaterials-13-00133-f005:**
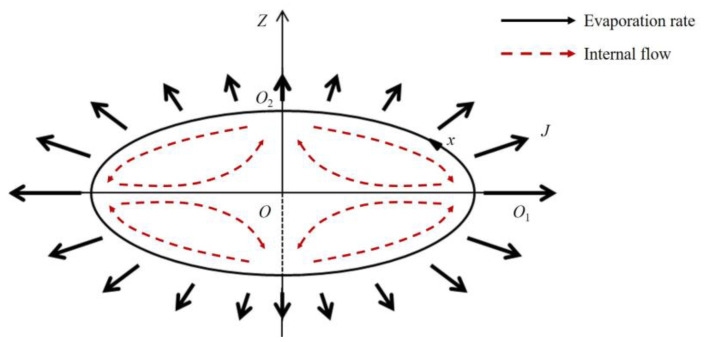
Distribution of the evaporation rate on the surface of an oblate acoustically levitated droplet. The arrows in the figure qualitatively represent the evaporation rate at local positions on the surface of the droplet and the longer the arrows are, the greater the evaporation rate is.

**Figure 6 nanomaterials-13-00133-f006:**
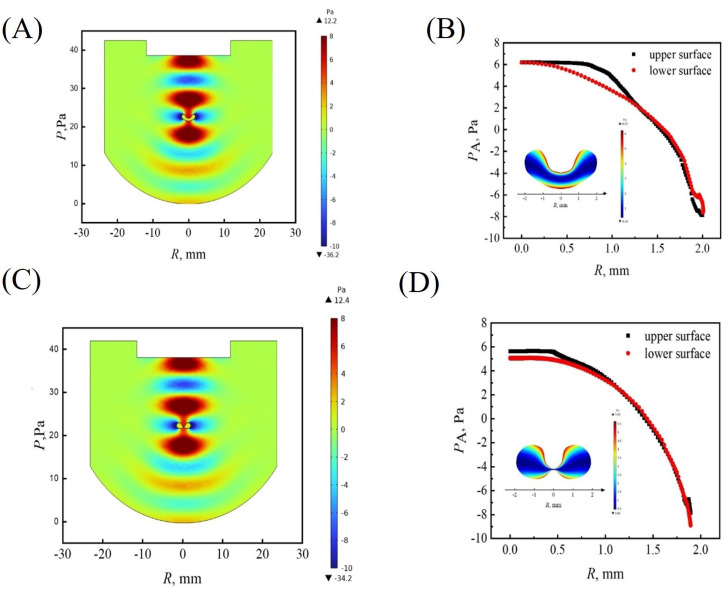
Acoustic radiation pressure in the sound field and on the surface of the final relic of the droplets with different morphology after solvent evaporation, (**A**) and (**B**), 60 wt% (**C**), and (**D**) 20 wt%.

**Table 1 nanomaterials-13-00133-t001:** Properties of the PTFE condensed solution used in the experiments.

Parameters	Value
PTFE solid content (wt%)	60 ± 2
Particle diameter (nm)	50
Kinematic viscosity (25 °C mm^2^/s)	6
Density (25 °C g/cm^3^)	1.50

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
