# Peer review of "Evaporation Caused Invaginations of Acoustically Levitated Colloidal Droplets"

_nanomaterials, 2022, doi:10.3390/nano13010133_

Round 1
Reviewer 1 Report
The authors submitted “Evaporation caused invagination of acoustically levitated colloidal droplets” for publication in “Nanomaterials”. The novelty of this work is good. Here are my comments.
1. English correction is needed.
2. the writing style should be improved, especially in the introduction part.
3. The figures should be combined together to reduce them.
4. The materials part should be 2.1.
5. Conclusion part is too lengthy.
Author Response
We thank the referee for his/her positive comments and the constructive suggestions for revision. We have made a comprehensive revision accordingly in the new version.
Reviewer 2 Report
This manuscript described the studies of the morphology evolution of PTFE colloidal droplets with different particle concentrations during evaporation under acoustic levitation. I think that this manuscript was well organized and what the authors intend to talk to readers was well explained with the various experimental data. The authors can review the manuscript carefully according to the following questions below.
1. In Table 1, the authors used 50 nm of particle diameter. Is there any difference of particle morphology when you use different particle diameter like less than or more than 50 nm.
Plus I am just wondering why the authors used 50 nm of particle diameter.
2. Could you explain the results of particle morphology according to the change of PTFE solid content?
3. I am wondering what the main driving force to make different particle morphology with changing the particle concentration.
Author Response
This manuscript described the studies of the morphology evolution of PTFE colloidal droplets with different particle concentrations during evaporation under acoustic levitation. I think that this manuscript was well organized and what the authors intend to talk to readers was well explained with the various experimental data. The authors can review the manuscript carefully according to the following questions below.
Response: We thank the referee for his/her positive comments.
- In Table 1, the authors used 50 nm of particle diameter. Is there any difference of particle morphology when you use different particle diameter like less than or more than 50 nm.
Plus I am just wondering why the authors used 50 nm of particle diameter.
Response: We thank the referee for the insightful question. To the best of our knowledge, similar researches have been carried out on colloidal droplets with particle size ranging from a few nanometers to tens of nanometers, but the drying structures are usually bowl-like or doughnut-like shape. Therefore, the size of particles is not the main factor affecting the buckling and invagination of the surface in the evaporation process, and we did not choose other sizes of particles for the experiment.
- Could you explain the results of particle morphology according to the change of PTFE solid content?
Response: Under the action of gravity, PTFE particles tend to deposit towards the bottom of the droplet. Therefore, the higher the concentration of PTFE, the more particles deposited at the bottom of the droplet, forming a thicker elastic shell that resists buckling. As a result, the drying structure changed from doughnut-like shape to bowl shape.
We have clarified this in the revised version, see p12-13.
- I am wondering what the main driving force to make different particle morphology with changing the particle concentration.
Response: The main driving force that changes the surface morphology of colloidal droplets during evaporation is the capillary force in the elastic shell and the acoustic radiation force on the droplet surface. Buckling occurs when the shell is too weak to resist capillary forces. Therefore, the drying structures formed by colloidal droplets with different particle concentrations are mainly caused by the difference in the thickness of elastic shell at the bottom of the dried droplet.
Round 2
Reviewer 1 Report
this manuscript is okay now for publication.